# A Comparative Assessment of the Upper Pharyngeal Airway Dimensions among Different Anteroposterior Skeletal Patterns in 7–14-Year-Old Children: A Cephalometric Study

**DOI:** 10.3390/children9081163

**Published:** 2022-08-03

**Authors:** Ann Chianchitlert, Suwannee Luppanapornlarp, Bhudsadee Saenghirunvattana, Irin Sirisoontorn

**Affiliations:** Department of Clinical Dentistry, Walailak University International College of Dentistry (WUICD), Walailak University, 87 Ranong 2 Road, Dusit, Bangkok 10300, Thailand; suwannee.lup@gmail.com (S.L.); ying.bsaeng@outlook.com (B.S.); irin.si@wu.ac.th (I.S.)

**Keywords:** pharyngeal airway, cephalometry, child development, skeletal pattern

## Abstract

Background: The pharyngeal airway is a crucial part of the respiratory system’s function. Assessing the pharyngeal airway dimensions in different skeletal types is important in the orthodontic treatment of growing patients. The aim of this study was to compare the upper pharyngeal airway dimensions of 7–14-year-old children with different skeletal types. Methods: Three-hundred-sixty-one lateral cephalometric radiographs were grouped based on their skeletal patterns determined by the ANB angle as skeletal type I (*n* = 123), type II (*n* = 121), and type III (*n* = 117). The radiographs were divided into 4 groups: 7/8 YO (7–8 years old), 9/10 YO, 11/12 YO, and 13/14 YO. The cephalometric measurements comprised SNA, SNB, ANB, Ad1-PNS, Ad2-PNS, McUP, and McLP. An ANOVA was used to compare the group results. Results: Significant differences in Ad1-PNS, Ad2-PNS, McUP, and McLP in skeletal types II and III were found between age groups. Most upper pharyngeal airway dimensions in skeletal types II and III children were significantly wider in the 13/14 YO group than in the other age groups. Conclusion: The upper pharyngeal airway dimensions increased age-dependently in 7–14-year-old children, especially in skeletal types II and III. The upper pharyngeal airway dimensions could serve as a guide in differentiating the different skeletal classes in clinical settings.

## 1. Introduction

Presently, there is evidence that reduced upper airway dimensions are associated with sleep breathing disorders [1]. This posterior airway space reduction is associated with the recurrence of obstructive sleep apnea (OSA) in adolescents [2]. The dental-related aspects of the posterior airway space should receive more consideration in pediatric OSA in growing individuals. Deformities of the craniofacial structures or soft tissues can be factors that reduce the pharyngeal space. Children with OSA syndrome are frequently found to have craniofacial anomalies. Children with simple snoring [3], adenotonsillar hypertrophy [4], obesity [5], and major craniofacial deformities (Pierre–Robin sequence or Down’s syndrome) [6] must be carefully evaluated for airway obstructions.

Previous studies have determined the dimensions of the upper pharyngeal airway in children [7,8]. However, these studies did not divide the children into age categories. Furthermore, few studies have examined the relationship between the upper airway dimensions and craniofacial structures using cephalometric assessments in growing individuals [9,10]. A previous study found a narrower nasopharyngeal airway in class II malocclusion subjects [11]. In preadolescents, volumetric measurements of the airway volume were correlated with the anterior facial height and ANB angle [12]. In contrast, Frietas et al. [13] found no association between the upper airway space and dental malocclusion types. The authors concluded that the malocclusion type did not affect the upper pharyngeal airway width. However, the pharyngeal airway volume was significantly correlated with the ANB angle and anterior facial height [12].

Previous studies have reported a relationship between the growth and development of the craniofacial features and pharyngeal airway [7,14]. It was also reported that the upper airway width increased with age; however, there were no significant differences between sexes in each age group [7]. In contrast, ethnic differences in upper pharyngeal airway dimensions have been reported [8,15,16,17]. Defining the normal values for the sagittal upper pharyngeal airway dimensions in children and adolescents is crucial for distinguishing deviations from the normative values that can allow for the early diagnosis of upper airway constriction [18,19].

Lateral cephalometric radiographs can be used in orthodontics to determine the craniofacial morphology and potential airway obstructions. It was reported that lateral cephalograms provide reliable nasopharyngeal dimension measurements [20]. Furthermore, there were no significant differences between using upright lateral cephalometric radiographs and supine position cone-beam computed tomography (CBCT) scans as screening techniques for detecting a constricted upper airway [21]. Various cephalometric variables are associated with the presence of OSA and its severity. The anatomic landmarks of the craniofacial structures and upper pharyngeal airway have been investigated for years [22]. Numerous cephalometric variables are associated with the presence of sleep disorder syndrome and its relative severity [23,24,25]. The results demonstrated that the anatomical landmarks of Ad1 (the intersection of the line from the posterior nasal spine and the adenoid outline) and Ad2 (the intersection of the line from the posterior nasal spine and the line from the midpoint of the line from the sella to basion points) were significantly correlated with the measured airway obstruction [24].

The upper pharyngeal airways in children with different skeletal types have been reported in many previous studies. Children with skeletal type II had significantly reduced dimensions of the inferior part of the upper pharyngeal airway [26]. In a study by Kim et al. [12], children with skeletal type II and retrusion of the mandible had narrower upper pharyngeal airway dimensions than skeletal type I individuals. In contrast, Gu et al. [8] reported a weak association between the upper pharyngeal airway and craniofacial structures in 12-year-old children. The growth and development of the upper pharyngeal airways in different skeletal types were investigated in several studies. The upper pharyngeal airway dimensions did not demonstrate age-related changes and were remarkably stable [14]. In contrast, Hsu et al. [27] reported that the pharyngeal airway dimensions were significantly correlated with age in 7–12-year-old children. However, a comparison of the development of the upper airway dimensions among these 3 skeletal types from children to adolescence has never been reported.

The previous studies led us to hypothesize that growth based on skeletal type differences might be a factor in increasing the dimensions of the pharyngeal airway. Therefore, the aim of this study was to evaluate the changes in the pharyngeal airway dimensions in 7–14-year-old children with skeletal types I, II, and III using a cephalometric analysis. In this study, we tested the null hypothesis that there were no differences in the upper pharyngeal airway dimensions among 7–14-year-old children with different skeletal types.

## 2. Materials and Methods

### 2.1. Ethics Approval

This retrospective study was approved by the Ethics Committee at Walailak University, Nakonsrithammarat, Thailand (WUEC 20-181-01). The samples were selected from the lateral cephalograms of children who presented for orthodontic consultation at a private dental clinic in Bangkok during 2014–2019.

### 2.2. Sample Size Calculation and Selection

The G*Power 3.1 package program analysis (G*Power, Heinrich-Heine-Universität Düsseldorf, Düsseldorf, Germany) was used to determine the required sample size [28]. When calculating the sample size, a type 1 error (α) = 0.05, effect size = 0.25, and test power (1-β) = 0.80 were used. The calculation indicated that a minimum of 60 patients per group (total sample size of 180 patients) would be necessary. Thus, 392 lateral cephalometric records were screened and 361 children (164 males, 197 females; mean age 10.87, S.D. = 2.23 years old) were selected for assessment based on the following inclusion criteria: healthy 7–14-year-old pediatric patients with no history of orthodontic treatment, no craniofacial abnormalities or genetic syndrome, no recent acute infection of the upper airway, no recent acute otitis media, no previous treatment for adenoidectomy or tonsillectomy, no temporomandibular joint disorders, and no serious systemic disease. The exclusion criteria were lateral cephalometric radiographs with mouth opening, swallowing, or tongue not in the rest position and poor clarity in the upper pharyngeal airway region. As categorized by ANB angle, the subjects consisted of 123 children with skeletal type I (34.1%), 121 children with skeletal type II (33.5%), and 117 children with skeletal type III (32.4%). The chronological age of the patient was used as a predictor of their growth. The age sub-classification was based on the lateral cephalometric radiographs taken 6 months before or after the subject’s birthday. Here, 87 children (23.4%) were in the 7/8 YO (7–8 years old) group, 92 children (25.5%) were in the 8/9 YO (8–9 years old) group, 95 children (24.3%) were in the 11/12 YO (11–12 years old) group, and 87 children (24.1%) were in the 13/14 YO (13–14 years old) group. The overall selection process is presented in Figure 1.

### 2.3. Lateral Cephalometric Radiograph Acquisition

The lateral cephalometric radiographs were taken using the manufacturers’ parameters. The Orthophos XG5DS (Sirona Dental Systems, Bensheim, Germany) cephalometric unit was used at settings of 63–69 kv and 8–12 mA. The cephalostat confirmed the natural head positioning, and the central beam was targeted at the left side of the face. The same 1.1 magnification factor was applied for all cephalometric radiographs. The radiographs were acquired with the subjects standing, with the head fixed to the cephalostat with ear rods and a support on the forehead. The teeth were in the maximum intercuspation, the lips were in a relaxed position, and the head was in a natural head position. The patients were instructed to bite in centric occlusion and to not swallow. The lateral cephalometric images’ contrast and brightness were manually adjusted, exported, and printed on paper using a 1:1 ratio in millimeters.

### 2.4. Anatomical Tracing and Measurement Variables

The images of the lateral cephalometric radiographs were hand-traced using a 0.3 mm lead pencil on 0.10 mm matte acetate tracing paper (G&H orthodontics, Franklin, IN, USA). One investigator traced and landmarked the craniofacial and upper pharyngeal airway structures by hand, as illustrated in Figure 2 [29,30]. The angle measurement was performed using a cephalometric protractor (Ormco, Glendora, CA, USA) with a 1:1 ratio in millimeters. The linear measurements were recorded using a Mitutoyo Digimatic caliper (Mituitoyo Corporation, Kawasaki-shi, Kanagawa, Japan). The intraexaminer reliability was determined by the intraclass correlation coefficient using an absolute agreement definition for each variable. The measurements were repeated in 20% of the radiographs 2 months later to calculate the intraexaminer reliability. The cephalometric analysis comprised common orthodontic angular (SNA, SNB, and ANB) and linear measurements and specific variables used for upper pharyngeal airway evaluation (Ad1-PNS, Ad2-PNS, McUP, and McLP) based on McNamara [29], Linder-Aronson [31], and Kirjavainen and Kirjavainen [30]. The ANB angle was used to determine the skeletal type for skeletal type I (2° ≤ ANB ≤ 5°), skeletal type II (ANB > 5°), and skeletal type III (ANB < 2°). The cephalometric variables used in previous pediatric obstructive apnea studies were chosen for this study [7,8,20,32,33].

### 2.5. Statistical Analysis

The data were analyzed using SPSS version 25 software (IBM SPSS, Armonk, NY, USA). The intraexaminer reliability was assessed using the intraclass correlation coefficient (ICC) for absolute agreement. Descriptive statistics including the means and standard deviations (SD) were calculated for all variables according to each skeletal type. The Kolmogorov–Smirnov and Shapiro–Wilk tests were performed to confirm the normal distribution of the data. The homogeneity of variance was assessed using Levene’s test. A one-way analysis of variance (ANOVA) followed by Dunnett’s test and a two-way ANOVA with Tukey’s test were used to test for the differences between groups where appropriate. Differences were considered significant when *p* ≤ 0.05.

## 3. Results

Because normal distribution and homogeneity of variance assumptions were met for our data, the means, SD, standard error of the mean (SEM), and parametric tests were used in this study. The ICC revealed high reliability for all lateral cephalometric measurements. The mean value for all measurements was 0.998 (minimum 0.995, maximum 0.999) for the intraobserver reliability. These ICC values indicated the consistent reproducibility of the measurements. The craniofacial morphologies as assessed by skeletal type and age group distribution are presented in Table 1. The demographic results showed that the age groups had a similar number of subjects, ranging from 87 to 95 subjects per group. Similarly, each skeletal type group comprised ~33% of the total subjects (*n* ranging from 117 to 123).

Changes in the upper pharyngeal airway dimensions from the 7–14-year-old children are presented in Table 2.

### 3.1. Cephalometric Analysis Data

Our results found that the SNA, SNB, and ANB values in skeletal types I and II were not significantly increased in any age group (*p* > 0.05) (Table 2). However, the SNB value was significantly increased in the 13/14 YO group with skeletal type III (*p* < 0.05). The ANB values were significantly different between skeletal types in all study age groups. In addition, a significant difference in SNB values was found between children with skeletal types II and III (*p* < 0.05). In contrast, there was no difference in SNA values between the three skeletal types in any age group.

### 3.2. Upper Pharyngeal Airway Dimensions

The ANOVA with Dunnett’s test revealed significant differences in Ad1-PNS, Ad2-PNS, and McUP, but not McLP, between the 7/8 YO and the other age groups in skeletal types II and III (*p* < 0.05). Based on these results, the null hypothesis that there were no differences in the upper pharyngeal airway dimensions among 7–14-year-old children with different skeletal types was rejected. Therefore, our findings suggest that the Ad1-PNS, Ad2-PNS, and McUP dimensions increased in an age-dependent manner (Table 2). There were no significant differences in Ad2-PNS, McUP, and McLP among the three skeletal types in the 7/8 YO group. However, the Ad1-PNS values were significantly different between skeletal types I and III in this age group. Furthermore, we found that the skeletal patterns did not affect most of the upper pharyngeal dimensions in any age group.

## 4. Discussion

The upper pharyngeal airway comprises several important structures that support the respiratory system’s function, i.e., the surrounding soft tissue conditions, such as the adenoids or tonsils, and the position of the maxilla and mandible. Therefore, it is important to investigate the age-related changes in upper airway dimensions between skeletal types I, II, and III children. The aim of this study was to use a lateral cephalometric analysis to define the changes in the upper pharyngeal airway dimensions throughout the growth range of 7–14-year-old children with skeletal types I, II, and III.

Because the vertical dimension impacts the A and B points, the ANB angle may not represent the actual anteroposterior relationship of the jaws [34]. However, the ANB angle is accepted to determine the anteroposterior skeletal pattern. The ANB angle was used to measure the relationship of the maxilla and mandible in this study. Although the homogeneity of the ANB angles in skeletal type III group can be critiqued, the age and ANB angle values in other skeletal types were comparable (Table 1).

As expected, our results showed a difference in ANB angles among the three skeletal types in growing children. Our findings are in line with those of Kim et al. [12] who reported a correlation between the upper airway volumes and ANB angles in preadolescents. In addition, the SNB values significantly increased from skeletal types II to III in all age groups. However, the skeletal patterns did not affect the SNA values in our study. Our results agreed with those of a previous study [35].

Our results found that the upper pharyngeal airway dimensions, Ad1-PNS and Ad2-PNS, increased with age-related growth, similar to a previous study [36]. It has been observed that the upper airway dimensions rapidly increase until the age of 13 [19]. During active growth, body maturity might be associated with the dimensions of the upper pharyngeal airway.

In our study, skeletal types II and III children showed age-dependent significant increases in their upper pharyngeal airway dimensions in almost all variables. However, the significantly narrower anteroposterior dimensions in the younger age group were not found in skeletal type I. The soft tissue structures surrounding the upper pharyngeal airway, such as the adenoids, may not have much of an effect on the upper pharyngeal airway in skeletal type I. Goncalves et al. [7] also found that the airway width increased with age; however, their study pooled all dental malocclusions together and more than half of the subjects were angle class II. In a study with a large sample size, Mislik et al. [14] reported that the mean upper airway dimensions remained stable between age groups. Our data suggest that the airway dimensions may be established at an early age and remain stable only in skeletal type I children.

Several studies have suggested relationships between the upper pharyngeal airway and anteroposterior skeletal pattern [12,37]. The skeletal class II children had smaller airway dimensions than those with skeletal classes I and III; however, these differences were not always significant [37]. Regarding the skeletal type, Alves et al. [38] found that the type of malocclusion did not affect most upper airway dimensions. In contrast, Kerr [39] revealed that class II malocclusion patients had narrower nasopharyngeal dimensions compared with class I patients. In the present study, 7-year-old children with skeletal type I had the widest upper pharyngeal airway dimensions. However, the upper pharyngeal airway dimensions were comparable in all skeletal types in 14-year-olds. Our results agreed with Gu et al. [8], who evaluated 12-year-old children and found weak associations between the upper airway and craniofacial structures. In the present study, neither significant differences between skeletal types nor age groups were found in terms of the SNA values. The anteroposterior position of the mandible, which was represented by the SNB, might affect the upper pharyngeal airway. Muto et al. [40] reported that the pharyngeal airway was larger with mandibular prognathism. The retrognathic mandible might be a factor for decreased dimensions of the upper pharyngeal airway.

In the present study, the upper pharyngeal airway dimensions significantly increased in the 10/11 YO and 13/14 YO groups compared with the 7/8 YO groups, which is the mandibular growth spurt phase according to Scammon’s growth curve [41]. Therefore, mandible growth may be a factor influencing the increase in upper pharyngeal airway dimensions. A future correlative longitudinal investigation of the upper pharyngeal airways of different skeletal patterns and mandibular growth will enhance our understanding of the relationship between mandibular growth and respiratory function.

In the present study, the increase in upper pharyngeal airway dimensions was age-dependent only in skeletal types II and III in 7–14-year-old children. At 7 years old, children with skeletal types II and III had narrower upper airway dimensions than those with skeletal type I. Because a skeletal discrepancy was also shown in the transverse dimension, a future study of the pharyngeal space should include a three-dimensional analysis. The early correction of skeletal types II and III might reduce the possibility of developing breathing disorders, such as snoring in children. Our results may be useful for diagnosis and treatment planning in orthodontics. Future studies should be undertaken to assess other contributing factors that potentially affect upper airway disorders.

In the present study, we collected the data from lateral cephalograms from an initial orthodontic screening phase from 2014 to 2019. The sample size used in this study exceeded the minimum requirements. We evaluated the changes in the upper pharyngeal airway dimensions in children with skeletal types I, II, and III throughout the growth of 7–14-year-olds using a lateral cephalometric analysis. The increase in upper pharyngeal airway dimensions was age-dependent only in skeletal types II and III. However, significantly increased dimensions were not found in skeletal type I. The upper pharyngeal airway dimensions significantly increased from 11 years old when compared with the 7/8 YO groups, which is the period of the mandibular growth spurt phase. Therefore, the mandible growth may be a factor that influences the increase in the upper pharyngeal airway dimensions.

Our results should be considered with caution. Because the upper pharyngeal airway involves complex anatomic structures, three-dimensional (3D) CBCT can optimally be used to provide a valuable three-dimensional analysis of the airways. If available, a CBCT examination of a pediatric patient should be performed when traditional radiography cannot provide an adequate diagnosis. Although using CBCT remains ideal, the child’s cooperation is required during CBCT scanning [42,43]. Lateral cephalograms have limited use in assessing the upper pharyngeal airway because they provide only a two-dimensional (2D) evaluation of the anteroposterior dimension. Moreover, the volume and thickness of the upper pharyngeal airway cannot be detected with a lateral cephalogram [44]. However, a study showed that measurements of the pharyngeal airway space using lateral cephalograms were similar to 3D-computed tomography (CT) scans, with a predictability rate of 92% [45]. Therefore, despite their limitations, lateral cephalograms can be a good supplementary tool because of their repeatability and reliability in evaluating the upper pharyngeal space [46].

Aboudara et al. [18] found a positive relationship between the linear dimensions from lateral cephalograms and the volume from CT scans of the upper pharyngeal airway in adolescents. Furthermore, Major et al. [47] suggested that cephalograms could be used as a screening tool for upper airway blockages, because they found a correlation between the linear measurements of the upper airway in a cephalogram. In addition, lateral cephalograms have been used as a supplementary imaging modality with a medical history for screening the constricted upper pharyngeal airway in pediatric OSA patients [48]. In the present study, lateral cephalograms were chosen to evaluate the upper pharyngeal airway dimensions because they are simple, readily available, and adhere to the as low as diagnostically acceptable principle of radiation for routine use for children [49]. Despite its shortcomings in diagnosis, cephalometry offers considerable advantages over other techniques, including being cost-effective and providing minimal exposure to radiation [18].

Taken together, our results suggest that growth is a factor in increasing the size of the upper pharyngeal airway dimensions. These dimensions increased in an age-dependent manner. In addition, the upper pharyngeal airway at Ad2-PNS in children with skeletal types II and III was significantly wider in the 13/14 YO group compared with the 7/8 YO group. However, we found that the SNB, but not SNA, was significantly increased in skeletal type III children. These results suggest that age has a marked influence on the mandible development compared with that of the maxilla.

## 5. Conclusions

In this study, we compared the upper pharyngeal airway dimensions of 7/8-, 9/10-, 11/12-, and 13/14-year-old children with skeletal types I, II, and III. Significant differences between age groups were found in terms of the Ad1-PNS, Ad2-PNS, McUP, and McLP values in skeletal types II and III. Therefore, this study suggests that the upper pharyngeal airway dimensions are related to age, especially in children with skeletal type III. Furthermore, we found that the SNB, but not SNA, was significantly increased in skeletal types II and III children. These results suggest that age has a marked influence on the mandible development compared with that of the maxilla.

## Figures and Tables

**Figure 1 children-09-01163-f001:**
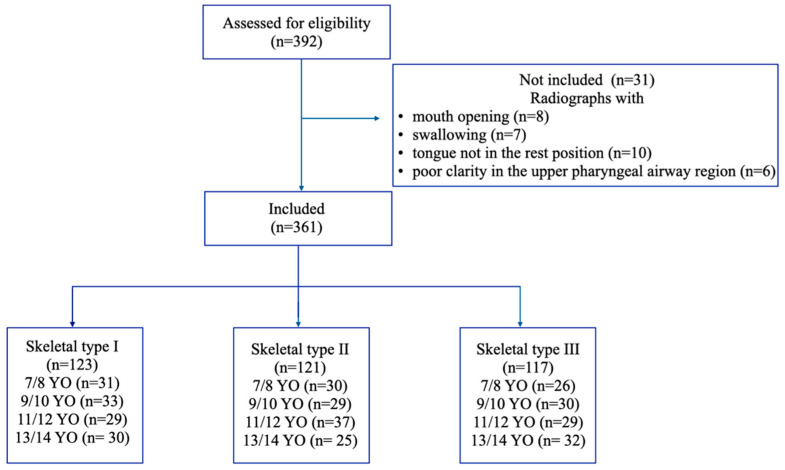
Flowchart of the record selection processes.

**Figure 2 children-09-01163-f002:**
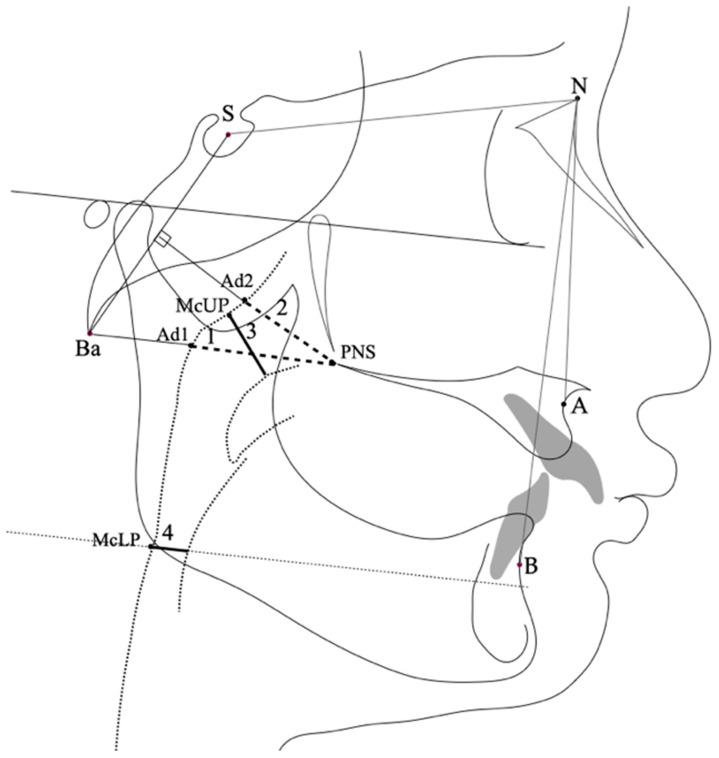
Reference points and lines on the cephalometric radiograph. Ba: basion; N: nasion; PNS: posterior nasal spine; S: sella; B: supramentale; A: subspinale. Craniofacial measurements: SNA: the angle between the sella (S), nasion (N), and A; SNB: the angle between the sella (S), nasion (N), and B; ANB: the angle between A, N, and B; Ad1-PNS (1): the distance of the intersection point of the posterior pharyngeal wall and the line from PNS to Ba; Ad2-PNS (2): the distance of the intersection point of the posterior pharyngeal wall and the line from the midpoint of the line from S to Ba; McUP (3): the shortest distance between the soft palate and the posterior pharyngeal wall; McLP (4): the shortest distance between the base of the tongue and the posterior pharyngeal wall.

**Table 1 children-09-01163-t001:** Distribution of ANB (°) values based on the subjects’ age and skeletal type.

	Skeletal Type I (*n* = 123) (M/F, 63:60)							
	7/8 YO (*n* = 31) (M/F, 15:16)		9/10 YO (*n* = 33) (M/F, 13:20)		11/12 YO (*n* = 29) (M/F, 16:13)		13/14 YO (*n* = 30) (M/F, 19:11)	
	Mean	SD	Mean	SD	Mean	SD	Mean	SD
Age (years)	7.77	0.43	9.63	0.33	11.78	0.44	13.59	0.50
ANB (°)	3.49	0.81	3.27	0.61	3.57	0.87	3.61	0.70
	Skeletal Type II (*n* = 121) (M/F, 41:80)							
	7/8 YO (*n* = 30) (M/F, 8:22)		9/10 YO (*n* = 29) (M/F, 13:16)		11/12 YO (*n* = 37) (M/F, 9:28)		13/14 YO (*n* = 25) (M/F, 11:14)	
	Mean	SD	Mean	SD	Mean	SD	Mean	SD
Age (years)	7.80	0.41	9.76	0.45	11.68	0.33	13.52	0.51
ANB (°)	6.42	1.39	6.28	1.24	6.19	0.91	6.08	0.87
	Skeletal Type III (*n* = 117) (M/F, 60:57)							
	7/8 YO (*n* = 26) (M/F, 17:9)		9/10 YO (*n* = 30) (M/F, 11:19)		11/12 YO (*n* = 29) (M/F, 15:14)		13/14 YO (*n* = 32) (M/F, 17:15)	
	Mean	SD	Mean	SD	Mean	SD	Mean	SD
Age (years)	7.58	0.50	9.59	0.58	11.49	0.32	13.69	0.47
ANB (°)	0.53	1.49	0.30	1.50	0.70	1.59	0.28	1.32

F, female; M, male.

**Table 2 children-09-01163-t002:** Comparison of the cephalometric variables and upper pharyngeal airway measurements among 7–14-year-old children in the three skeletal types.

		7/8 YO		9/10 YO		11/12 YO		13/14 YO	
		Mean	SEM	Mean	SEM	Mean	SEM	Mean	SEM
SNA (°)	Skeletal type I	81.21 ^A,B a^ (*n* = 31)	0.75	80.74 ^A a^ (*n* = 33)	0.70	81.02 ^A a^ (*n* = 29)	0.53	83.01 ^A a^ (*n* = 30)	0.71
	Skeletal type II	82.31 ^B a^ (*n* = 30)	0.61	82.13 ^A a^ (*n* = 29)	0.66	82.46 ^A a^ (*n* = 37)	0.49	83.90 ^A a^ (*n* = 25)	0.31
	Skeletal type III	79.96 ^A a^ (*n* = 26)	0.76	81.16 ^A a^ (*n* = 30)	0.48	80.39 ^A a^ (*n* = 29)	0.66	82.18 ^A a^ (*n* = 32)	0.61
SNB (°)	Skeletal type I	77.74 ^A a^ (*n* = 31)	0.73	77.49 ^A a^ (*n* = 33)	0.66	77.44 ^A a^ (*n* = 29)	0.49	79.40 ^A a^ (*n* = 30)	0.68
	Skeletal type II	75.82 ^B a^ (*n* = 30)	0.64	75.86 ^A a^ (*n* = 29)	0.61	76.26 ^A a^ (*n* = 37)	0.51	77.57 ^A a^ (*n* = 25)	0.43
	Skeletal type III	79.47 ^B a^ (*n* = 26)	0.75	80.86 ^B a^ (*n* = 30)	0.53	79.66 ^B a^ (*n* = 29)	0.77	81.83 ^B b^ (*n* = 32)	0.64
ANB (°)	Skeletal type I	3.49 ^A a^ (*n* = 31)	0.15	3.27 ^A a^ (*n* = 33)	0.11	3.57 ^A a^ (*n* = 29)	0.16	3.61 ^A a^ (*n* = 30)	0.13
	Skeletal type II	6.42 ^B a^ (*n* = 30)	0.25	6.28 ^B a^ (*n* = 29)	0.23	6.19 ^B a^ (*n* = 37)	0.15	6.08 ^B a^ (*n* = 25)	0.17
	Skeletal type III	0.53 ^C a^ (*n* = 26)	0.29	0.30 ^C a^ (*n* = 30)	0.27	0.70 ^C a^ (*n* = 29)	0.30	0.28 ^C a^ (*n* = 32)	0.23
Ad1-PNS (mm)	Skeletal type I	19.29 ^A a^ (*n* = 31)	0.56	19.82 ^A a^ (*n* = 33)	0.66	20.67 ^A a^ (*n* = 29)	0.89	20.64 ^A a^ (*n* = 30)	0.69
	Skeletal type II	18.02 ^A,B a^ (*n* = 30)	0.77	18.79 ^A a^ (*n* = 29)	0.80	18.52 ^A a^ (*n* = 37)	0.61	21.33 ^A b^ (*n* = 25)	0.52
	Skeletal type III	16.81 ^B a^ (*n* = 26)	0.98	19.81 ^A b^ (*n* = 30)	0.62	21.55 ^A b^ (*n* = 29)	0.64	21.46 ^A b^ (*n* = 32)	0.79
Ad2-PNS (mm)	Skeletal type I	13.51 ^A a^ (*n* = 31)	0.45	14.02 ^A a^ (*n* = 33)	0.58	15.11 ^A a^ (*n* = 29)	0.70	15.63 ^A a^ (*n* = 30)	0.71
	Skeletal type II	12.06 ^A a^ (*n* = 30)	0.44	13.06 ^A a^ (*n* = 29)	0.72	14.12 ^A b^ (*n* = 37)	0.56	16.44 ^A b^ (*n* = 25)	0.47
	Skeletal type III	12.23 ^A a^ (*n* = 26)	0.82	14.04 ^A a^ (*n* = 30)	0.55	16.10 ^A b^ (*n* = 29)	0.89	16.82 ^A b^ (*n* = 32)	0.65
McUP (mm)	Skeletal type I	9.40 ^A a^ (*n* = 31)	0.38	10.39 ^A a^ (*n* = 33)	0.60	10.39 ^A a^ (*n* = 29)	0.75	11.45 ^A a^ (*n* = 30)	0.74
	Skeletal type II	7.72 ^A a^ (*n* = 30)	0.39	8.16 ^B a^ (*n* = 29)	0.50	9.99 ^A b^ (*n* = 37)	0.54	12.01 ^A b^ (*n* = 25)	0.52
	Skeletal type III	8.80 ^A a^ (*n* = 26)	0.71	9.84 ^A a^ (*n* = 30)	0.58	11.74 ^A b^ (*n* = 29)	0.71	12.89 ^A b^ (*n* = 32)	0.67
McLP (mm)	Skeletal type I	9.07 ^A a^ (*n* = 31)	0.54	10.15 ^A a^ (*n* = 33)	0.54	10.33 ^A a^ (*n* = 29)	0.69	10.07 ^A a^ (*n* = 30)	0.04
	Skeletal type II	9.90 ^A a^ (*n* = 30)	0.53	9.03 ^A,B a^ (*n* = 29)	0.50	9.44 ^A a^ (*n* = 37)	0.41	9.970 ^A a^ (*n* = 25)	0.78
	Skeletal type III	9.90 ^A a^ (*n* = 26)	0.58	9.65 ^B a^ (*n* = 30)	0.52	10.05 ^A a^ (*n* = 29)	0.47	11.59 ^A b^ (*n* = 32)	0.55

^A–C^ Values with different capital superscript letters in a column indicate significant differences after a two-way ANOVA and Tukey’s test (*p* ≤ 0.05). ^a,b^ Values with different lowercase superscript letters in a row are significant compared with the corresponding 7/8 YO group after a one-way ANOVA and Dunnett’s test (*p* ≤ 0.05).

## Data Availability

Not applicable.

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
