# Peer review of "A Comparative Assessment of the Upper Pharyngeal Airway Dimensions among Different Anteroposterior Skeletal Patterns in 7–14-Year-Old Children: A Cephalometric Study"

_children, 2022, doi:10.3390/children9081163_

Round 1

Reviewer 1 Report

Dear Authors, congratulation on your work, i believe is a good study, i would like to point out somethings in order to improve your article:

1.-The introduction should be increased in order to give a state of the art of the aim of the study. You should emphasys more on the anatomic side of the study in the introduction. 

2.- You should write on the merial and methods the ethic commitee you passed in order to do the study.

3.- You shold review the editing of the article:

figure 1

figure 2

4.- In results yoou have a lot of data, you should present the data that is important to the study.

5.- You should consider writting the conclusions with a structure accordin gthe aims of the study.

Author Response

Response to reviewers

We thank the reviewer for the careful reading of our manuscript and giving suggestions that have allowed us to improve the revised version.  The revised passages based on the reviewer’s comments are highlighted in yellow in the re-submitted manuscript. Additional revised passages are highlighted in blue.  Our responses to the comments are found below:

Response to Reviewer 1 Comments

Dear Authors, congratulation on your work, I believe is a good study, I would like to point out somethings in order to improve your article:

Thank you for your positive comments and kind suggestions.

  1. The introduction should be increased in order to give a state of the art of the aim of the study. You should emphasys more on the anatomic side of the study in the introduction.

Author response: We agree with the reviewer to emphasize more on the anatomic aspect of the study in the Introduction on page 3.

Revised Text:

The anatomic landmarks of the craniofacial structures and upper pharyngeal airway have been investigated for years [22]. Numerous cephalometric variables are associated with the presence of sleep disordered syndrome and its relative severity [23-25]. The results demonstrated that the anatomical landmarks of Ad1 (intersection of the line from the Posterior nasal spine and the adenoid outline) and Ad2 (intersection of the line from the posterior nasal spine and the line from the midpoint of the line from Sella to Basion) were significantly correlated with the measured airway obstruction [24].

  1. You should write on the material and methods the ethic commitee you passed in order to do the study.

Author response: The ethics committee and approval number were included in the materials and methods on page 3.

Revised text:

This retrospective study was approved by the Ethics Committee at Walailak University, Nakonsrithammarat, Thailand (WUEC 20-181-01).

3.-You should review the editing of the article:

Figure 1

Figure 2

Author response: Thanks for your careful reading. The Figure 1 and its legend were revised on Page 5. The Figure 2 and its legend were revised on page 6. 

Revised text:

Figure 1. Flowchart of the record selection processes.

Figure 2. Reference points and lines on the cephalometric radiograph. Ba: basion; N: nasion; PNS: posterior nasal spine; S: sella; B: supramentale; A: subspinale; Craniofacial measurements—SNA: The angle between sella (S), nasion (N) and A; SNB: The angle between sella (S), nasion (N) and B; ANB: The angle between A, N, and B; Ad1-PNS (1): the distance of the intersection point of posterior pharyngeal wall and the line from PNS to Ba; Ad2-PNS (2): the distance of the intersection point of posterior pharyngeal wall and the line from the midpoint of the line from S to Ba; McUP (3): The shortest distance between the soft palate and the posterior pharyngeal wall; McLP (4): The shortest distance between the base of the tongue and the posterior pharyngeal wall.

4.- In results you have a lot of data, you should present the data that is important to the study.

Author response: To highlight the important results, we merged Tables 2 and 3 to be the new Table 2 in the revised manuscript on page 9.  Figures 3 and 4 which were redundant to the new Table 2 were deleted.  The revised important results in text is on page 10-11.

Revised text

Table 2. Comparison of the cephalometric variables and upper pharyngeal airway measurements among 7–14-year-old children in the three skeletal types.

7/8 YO

9/10 YO

11/12 YO

13/14 YO

Mean

SEM

Mean

SEM

Mean

SEM

Mean

SEM

SNA (o)

Skeletal type I

81.21 A,B a (n=31)

0.75

80.74 A a (n=33)

0.70

81.02 A a (n=29)

0.53

83.01 A a (n=30)

0.71

Skeletal type II

82.31 B a (n=30)

0.61

82.13 A a (n=29)

0.66

82.46 A a (n=37)

0.49

83.90 A a (n=25)

0.31

Skeletal type III

79.96 A a (n=26)

0.76

81.16 A a (n=30)

0.48

80.39 A a (n=29)

0.66

82.18 A a (n=32)

0.61

SNB (o)

Skeletal type I

77.74 A a (n=31)

0.73

77.49 A a (n=33)

0.66

77.44 A a (n=29)

0.49

79.40 A a (n=30)

0.68

Skeletal type II

75.82 B a (n=30)

0.64

75.86 A a (n=29)

0.61

76.26 A a (n=37)

0.51

77.57 A a (n=25)

0.43

Skeletal type III

79.47 B a (n=26)

0.75

80.86 B a (n=30)

0.53

79.66 B a (n=29)

0.77

81.83 B b (n=32)

0.64

ANB (o)

Skeletal type I

3.49 A a (n=31)

0.15

3.27 A a (n=33)

0.11

3.57 A a (n=29)

0.16

3.61 A a (n=30)

0.13

Skeletal type II

6.42 B a (n=30)

0.25

6.28 B a (n=29)

0.23

6.19 B a (n=37)

0.15

6.08 B a (n=25)

0.17

Skeletal type III

0.53 C a (n=26)

0.29

0.30 C a (n=30)

0.27

0.70 C a (n=29)

0.30

0.28 C a (n=32)

0.23

Ad1-PNS (mm)

Skeletal type I

19.29A a (n=31)

0.56

19.82 A a (n=33)

0.66

20.67 A a (n=29)

0.89

20.64 A a (n=30)

0.69

Skeletal type II

18.02 A,B a (n=30)

0.77

18.79 A a (n=29)

0.80

18.52 A a (n=37)

0.61

21.33 A b (n=25)

0.52

Skeletal type III

16.81 B a (n=26)

0.98

19.81 A b (n=30)

0.62

21.55 A b (n=29)

0.64

21.46 A b (n=32)

0.79

Ad2-PNS (mm)

Skeletal type I

13.51 A a (n=31)

0.45

14.02 A a (n=33)

0.58

15.11 A a (n=29)

0.70

15.63 A a (n=30)

0.71

Skeletal type II

12.06 A a (n=30)

0.44

13.06 A a (n=29)

0.72

14.12 A b (n=37)

0.56

16.44 A b (n=25)

0.47

Skeletal type III

12.23 A a (n=26)

0.82

14.04 A a (n=30)

0.55

16.10 A b (n=29)

0.89

16.82 A b (n=32)

0.65

McUP (mm)

Skeletal type I

9.40 A a (n=31)

0.38

10.39 A a (n=33)

0.60

10.39 A a (n=29)

0.75

11.45 A a (n=30)

0.74

Skeletal type II

7.72 A a (n=30)

0.39

8.16 B a (n=29)

0.50

9.99 A b (n=37)

0.54

12.01 A b (n=25)

0.52

Skeletal type III

8.80 A a (n=26)

0.71

9.84 A a (n=30)

0.58

11.74 A b (n=29)

0.71

12.89 A b (n=32)

0.67

McLP (mm)

Skeletal type I

9.07 A a (n=31)

0.54

10.15 A a (n=33)

0.54

10.33 A a (n=29)

0.69

10.07 A a (n=30)

.043

Skeletal type II

9.90 A a (n=30)

0.53

9.03 A,B a (n=29)

0.50

9.44 A a (n=37)

0.41

9.970 A a (n=25)

0.78

Skeletal type III

9.90 A a (n=26)

0.58

9.65 B a (n=30)

0.52

10.05 A a (n=29)

0.47

11.59 A b (n=32)

0.55

A-C Values with different capital superscript letters in a column show significant difference after a two-way ANOVA and Turkey’s test (P ≤ 0.05).

a,b Values with different small superscript letters in a row are significantly compared with the corresponding 7/8 YO group after a one-way ANOVA and Dunnett’ s test (P ≤  0.05).

3.1. Cephalometric analysis data.

Our results demonstrated that SNA, SNB, and ANB in skeletal types I and II were not significantly increased in all age groups (P > 0.05) (Table). However, SNB was significantly increased in the 13/14YO group with skeletal type III (P < 0.05). ANB was significantly different between skeletal types in all study age groups. In addition, a significant difference in SNB was found between children with skeletal type II and III (P < 0.05). In contrast, there was no difference in SNA between the three skeletal types in all age groups.

3.2. Upper pharyngeal airway dimensions

ANOVA with Dunnett’s test revealed significant differences in Ad1-PNS, Ad2-PNS, and McUP, but not McLP, between the 7/8 YO and the other age groups in skeletal type II and III (p < 0.05). The Ad1-PNS, Ad2-PNS, and McUP dimensions increased in an age-dependent manner (Table 2). There was no significant difference in Ad2-PNS, McUP, and McLP among three skeletal types in the 7/8 YO group. However, Ad1-PNS was significantly different between skeletal type I and III in this age group. Furthermore, we found that skeletal patterns did not affect most of the upper pharyngeal dimensions in all age groups.

5.- You should consider writting the conclusions with a structure according the aims of the study.

Author response: We have updated the conclusions with a structure according to the aim of the study on page 13.

Revised text:

Significant differences between age groups were found in Ad1-PNS, Ad2-PNS, McUP, and McLP in skeletal types II and III. Therefore, this study demonstrates that the upper pharyngeal airway dimensions are related to age, especially in children with skeletal type II and III. Furthermore, we found that SNB, but not SNA, significantly increased in skeletal type II and III children. These results suggest that age has a marked influence on mandible development compared with that of the maxilla.

Reviewer 2 Report

The work is very well structured and executed but there are some criticisms to be solved:

-Check that all keywords are MESH terms from the Pubmed database

-In the abstract section, insert an opening sentence about the generic problem on the scaffolds

-Insert instead a final sentence on the possible clinical repercussions of the study

-At the end of the introduction section, insert the null hypotheses of the study which must be refuted at the end of the discussion section

- Enter the identification number of the relevant ethics committee

-Insert the reference details of the G power 3.1 program

-All the figures appear not paginated correctly. I consider this a lack of respect for the audit work

-Some general considerations should be included on the relationship between orofacial dimensions and specific syndromes. In this regard, I ask that the following scientific work be included in the reference section, which could be of help to the reader:

- Giuca MR, Carli E, Lardani L, Pasini M, Miceli M, Fambrini E. Pediatric Obstructive Sleep Apnea Syndrome: Emerging Evidence and Treatment Approach. ScientificWorldJournal. 2021; 2021: 5591251. Published 2021 Apr 23. doi: 10.1155 / 2021/5591251

Author Response

Response to reviewers

We thank the reviewer for the careful reading of our manuscript and giving suggestions that have allowed us to improve the revised version.  The revised passages based on the reviewer’s comments are highlighted in yellow in the re-submitted manuscript. Additional revised passages are highlighted in blue.  Our responses to the comments are found below:

Response to Reviewer 2 Comments

The work is very well structured and executed but there are some criticisms to be solved:

Thank you for your positive comments and kind suggestions.  We apologize for the incorrectly paginated figures. Thank you for your careful reading.

  • Check that all keywords are MESH terms from the Pubmed database

Author response: The keywords were revised on page 2.

Revised text: pharyngeal airway; cephalometry; child development; skeletal pattern

  • In the abstract section, insert an opening sentence about the generic problem on the scaffolds

Author response: An opening sentence about the generic problem was inserted in the abstract on page 2.

Revised text:

The pharyngeal airway is a crucial part of the respiratory system’s function. Assessing the pharyngeal airway dimensions in different skeletal types is important in the orthodontic treatment of growing patients.

  • Insert instead a final sentence on the possible clinical repercussions of the study

Author response: A final sentence on the possible clinical repercussion was inserted in the abstract on page 2.

Revised text:

The upper pharyngeal airway dimensions could serve as a guide in differentiating the different skeletal classes in clinical settings.

  • At the end of the introduction section, insert the null hypotheses of the study which must be refuted at the end of the discussion section

Author response: The null hypotheses was inserted in the end of the introduction section on page 3.  The rejection of the null hypothesis was inserted at the end of the discussion on page 13.

Revised text:

Introduction

The null hypothesis was that there were no differences in upper pharyngeal airway dimensions among 7-14-year-old children with different skeletal types.

Discussion

Our results support the hypothesis that growth is a factor in increasing the size of the upper pharyngeal airway dimensions. These dimensions increased in an age-dependent manner. In addition, the upper pharyngeal airway at Ad2-PNS in children with skeletal type II and III were significantly wider in the 13/14 YO group compared with the 7/8 YO group. Based on these results, the null hypothesis was rejected.

  • Enter the identification number of the relevant ethics committee

Author response: The identification number of the relevant ethics committee was revised in the new manuscript on page 3.

Revised text:

This retrospective study was approved by the Ethics Committee at Walailak University, Nakonsrithammarat, Thailand (WUEC 20-181-01).

  • Insert the reference details of the G power 3.1 program

Author response: The reference details including a related reference of the G power 3.1 program were inserted on page 4.

Revised text:

The G*Power 3.1 package program analysis G*Power, Heinrich-Heine-Universität Düsseldorf, Düsseldorf, Germany) was used to determine the required sample size [28].

Reference: Faul, F.; Erdfelder, E.; Lang, A.-G.; Buchner, A. G*Power 3: A flexible statistical power analysis program for the social, behavior, and biomedical sciences. Behav Res Meth Instrum & Comput 2007, 39, 175-191, doi:10.3758/BF03193146.

  • All the figures appear not paginated correctly. I consider this a lack of respect for the audit work

Author response: We apologize for this inadvertent writing error. The errors in Figure 1 were corrected.  The legend of Figure 1 was rewritten (page 5). We also corrected the legend of Figure 2 on page 6.  Furthermore, Figures 3 and 4 in the old manuscript were deleted due to their redundant data with the new Table 2 in the revised manuscript.

Revised text:

Figure 1. Flowchart of the record selection processes.

Figure 2. Reference points and lines on the cephalometric radiograph. Ba: basion; N: nasion; PNS: posterior nasal spine; S: sella; B: supramentale; A: subspinale; Craniofacial measurements—SNA: The angle between sella (S), nasion (N) and A; SNB: The angle between sella (S), nasion (N) and B; ANB: The angle between A, N, and B; Ad1-PNS (1): the distance of the intersection point of posterior pharyngeal wall and the line from PNS to Ba; Ad2-PNS (2): the distance of the intersection point of posterior pharyngeal wall and the line from the midpoint of the line from S to Ba; McUP (3): The shortest distance between the soft palate and the posterior pharyngeal wall; McLP (4): The shortest distance between the base of the tongue and the posterior pharyngeal wall.

  • Some general considerations should be included on the relationship between orofacial dimensions and specific syndromes. In this regard, I ask that the following scientific work be included in the reference section, which could be of help to the reader:

- Giuca MR, Carli E, Lardani L, Pasini M, Miceli M, Fambrini E. Pediatric Obstructive Sleep Apnea Syndrome: Emerging Evidence and Treatment Approach. ScientificWorldJournal. 2021; 2021: 5591251. Published 2021 Apr 23. doi: 10.1155 / 2021/5591251

Author response: We thank the reviewer for the suggestion of a related reference involving the general considerations of specific syndromes.  In the revised manuscript, we added the reference as such (reference 6).  In addition, the other references related to adenotonsillar hypertrophy (reference 4), obesity (reference 5), and major craniofacial deformities (Pierre Robin sequence or Down Syndrome) (reference 6) were included in the introduction part of the revised manuscript. (Page 2)

Revised text:

Deformities of the craniofacial structures or soft tissues can be factors that reduce the pharyngeal space. Children with OSA syndrome are frequently found to have craniofacial anomalies. Children with simple snoring [3], adenotonsillar hypertrophy [4], obesity [5], and major craniofacial deformities (Pierre Robin sequence or Down Syndrome) [6] must be carefully evaluated for airway obstruction.

Added References

  1. Savini, S.; Ciorba, A.; Bianchini, C.; Stomeo, F.; Corazzi, V.; Vicini, C.; Pelucchi, S. Assessment of obstructive sleep apnoea (OSA) in children: an update. Acta Otorhinolaryngol Ital 2019, 39, 289-297, doi:10.14639/0392-100x-n0262.
  2. Suen, J.S.; Arnold, J.E.; Brooks, L.J. Adenotonsillectomy for Treatment of Obstructive Sleep Apnea in Children. Archives of Otolaryngology–Head & Neck Surgery 1995, 121, 525-530, doi:10.1001/archotol.1995.01890050023005.
  3. Arens, R.; Muzumdar, H. Childhood obesity and obstructive sleep apnea syndrome. J Appl Physiol 2010, 108, 436-444, doi:10.1152/japplphysiol.00689.2009.
  4. Giuca, M.R.; Carli, E.; Lardani, L.; Pasini, M.; Miceli, M.; Fambrini, E. Pediatric Obstructive Sleep Apnea Syndrome: Emerging Evidence and Treatment Approach. Sci World J 2021, 2021, 5591251, doi:10.1155/2021/5591251.

Round 2

Reviewer 1 Report

Dear authors I have seen an improvement on your article, congratulations. I have to point out some details that still have to be changed but it is a better article now. 
You should improve results, you can’t say this article demonstrates, because you don’t have the data to demonstrate, you can say that the data in this study suggest, but nothing more.

The discussion  can not finish with the sentence the null hipotheses has to be rejected, that is part of the results.

With minor changes I hope the article could be more improved and also have the chance to be published 

Author Response

Response to Reviewer 1 Comments

Response to Reviewer 1 Comments

Dear authors I have seen an improvement on your article, congratulations. I have to point out some details that still have to be changed but it is a better article now. 

Thank you for your positive comments and kind suggestions.

( ) Extensive editing of English language and style required  
( ) Moderate English changes required  
(x) English language and style are fine/minor spell check required  
( ) I don't feel qualified to judge about the English language and style 

Does the introduction provide sufficient background and include all relevant references?

 Yes

(x)

Can be

improved

( )

Must be improved

( )

Not applicable

( )

Are all the cited references relevant to the research?

(x)

( )

( )

( )

Is the research design appropriate?

(x)

( )

( )

( )

Are the methods adequately described?

( )

(x)

( )

( )

Are the results clearly presented?

( )

(x)

( )

( )

Are the conclusions supported by the results?

( )

( )

(x)

( )

Point 1: You should improve results, you can’t say this article demonstrates, because you don’t have the data to demonstrate, you can say that the data in this study suggest, but nothing more.

Response 1: The results were improved. Multiple sentences were revised to replace ‘demonstrate’. (line 245-247 and 304-305)

Revised text:

Results

            Because a normal distribution and homogeneity of variance assumptions were met for our data, means, SD, standard error of the mean (SEM), and parametric testes were used in this study. (line 245-247)

            Our results found that SNA, SNB, and ANB in skeletal types I and II were not significantly increased in all age groups (P > 0.05) (Table 2). (line 304-305)

Point 2: The discussion can not finish with the sentence the null hipotheses has to be rejected, that is part of the results.

Response 2: The sentence of the null hypothesis was placed in the results section. (line 314-316)

Revised text:

            Based on these results, the null hypothesis that there were no differences in upper pharyngeal airway dimensions among 7–14-year-old children with different skeletal types was rejected.  Therefore, our findings suggest that the Ad1-PNS, Ad2-PNS, and McUP dimensions increased in an age-dependent manner (Table 2).

Point 3: Are the methods adequately described?

Response 3: Additional method details were included. (line 146-201)

Revised text:

            2.3 Lateral cephalometric radiograph acquisition

The lateral cephalometric radiographs were taken using the manufacturers’ parameters. The Orthophos XG5DS (Sirona Dental Systems, Bensheim, Germany) cephalometric unit was used at settings of 63-69 kv, and 8-12 mA. The cephalostat confirmed natural head positioning, and the central beam was targeted at the left side of the face.  The same 1.1 magnification factor was applied for all cephalometric radiographs.  The radiographs were acquired with the subjects standing, the head was fixed to the cephalostat with ear rods and a support on the forehead. The teeth were in the maximum intercuspation, the lips were in a relaxed position, and the head was in a natural head position. The patients were instructed to bite in centric occlusion and to not swallow. The lateral cephalometric images’ contrast and brightness were manually adjusted, exported, and printed on paper using a 1:1 ratio in millimeters.

  2.4 Anatomical Tracing and Measurement Variables

The images of the lateral cephalometric radiographs were hand-traced using a 0.3 mm lead pencil on 0.10 mm matte acetate tracing paper (G&H orthodontics, Franklin, Indiana, USA). One investigator traced and landmarked the craniofacial and upper pharyngeal airway structures by hand as illustrated in Figure 2 [29,30]. The angle measurement was done using a cephalometric protractor (Ormco, Glendora, California, USA) with a 1:1 ratio in millimeters. The linear measurements were recorded using a Mitutoyo Digimatic caliper (Mituitoyo Corporation, Kawasaki-shi, Kanagawa, Japan). The intra-examiner reliability was determined by the intraclass correlation coefficient using an absolute agreement definition for each variable. The measurements were repeated in 20% of the radiographs 2 months later to calculate the intra-examiner reliability. The cephalometric analysis comprised common orthodontic angular (SNA, SNB, and ANB) and linear measurements and specific variables used for upper pharyngeal airway evaluation (Ad1-PNS, Ad2-PNS, McUP, and McLP) based on McNamara [29], Linder-Aronson [31], and Kirjavainen and Kirjavainen [30]. The ANB angle was used to determine the skeletal type; Skeletal type I (2°≤ANB ≤5°), Skeletal type II (ANB >5°), and Skeletal type III (ANB<2°).  The cephalometric variables used in previous pediatric obstructive apnea studies were chosen for this study [7,8,20,32,33].

Point 4: Are the results clearly presented?

Response 4: The results were more clearly presented.(line 314-317)

Revised text:

            Based on these results, the null hypothesis that there were no differences in upper pharyngeal airway dimensions among 7–14-year-old children with different skeletal types was rejected. Therefore, our findings suggest that the Ad1-PNS, Ad2-PNS, and McUP dimensions increased in an age-dependent manner (Table 2).

Point 5: Are the conclusions supported by the results?

Response 5: The conclusions were revised to be supported by the results. (line 448-455)

Revised text:

            In this study, we compared the upper pharyngeal airway dimensions of 7/8-, 9/10-, 11/12-, and 13/14-year-old children with skeletal types I, II, and III. Significant differences between age groups were found in Ad1-PNS, Ad2-PNS, McUP, and McLP in skeletal types II and III. Therefore, this study suggests that the upper pharyngeal airway dimensions are related to age, especially in children with skeletal type III. Furthermore, we found that SNB, but not SNA, significantly increased in skeletal type II and III children. These results suggest that age has a marked influence on mandible development compared with that of the maxilla.

Reviewer 2 Report

All comments were added 

Author Response

Response to Reviewer 2 Comments

Thanks for your positive comments and suggestions.

( ) Extensive editing of English language and style required  
(x) Moderate English changes required  
( ) English language and style are fine/minor spell check required  
( ) I don't feel qualified to judge about the English language and style 

Does the introduction provide sufficient background and include all relevant references?

 Yes

(x)

Can be

improved

( )

Must be improved

( )

Not applicable

( )

Are all the cited references relevant to the research?

(x)

( )

( )

( )

Is the research design appropriate?

(x)

( )

( )

( )

Are the methods adequately described?

(x)

( )

( )

( )

Are the results clearly presented?

(x)

( )

( )

( )

Are the conclusions supported by the results?

(x)

( )

( )

( )

Point 1: Moderate English changes required.

Response 1: The revised manuscript has been edited and reviewed for English language by a native speaker, Dr. Kevin Tompkins (Associate Editor of Connective Tissue Research).
